# A Planar Disk Electrode Chip Based on MWCNT/CS/Pb^2+^ Ionophore IV Nanomaterial Membrane for Trace Level Pb^2+^ Detection

**DOI:** 10.3390/molecules28104142

**Published:** 2023-05-17

**Authors:** Yuan Zhuang, Cong Wang, Wei Qu, Yirou Yan, Ping Wang, Chengjun Qiu

**Affiliations:** 1College of Mechanical, Naval Architecture & Ocean Engineering, Beibu Gulf University, Qinzhou 535011, China; 2College of Electronics and Information Engineering, Beibu Gulf University, Qinzhou 535011, China; 3Guangxi Key Laboratory of Ocean Engineering Equipment and Technology, Qinzhou 535011, China

**Keywords:** multiwalled carbon nanotube, chitosan, lead ionophore IV, planar disk electrode

## Abstract

Unlike conventional lead ion (Pb${}^{2+}$) detecting methods, electrochemical methods have the attractive advantages of rapid response, good portability and high sensitivity. In this paper, a planar disk electrode modified by multiwalled carbon nanotube (MWCNTs)/chitosan (CS)/lead (Pb${}^{2+}$) ionophore IV nanomaterial and its matched system are proposed. This system presented a good linear relationship between the concentration of Pb${}^{2+}$ ions and the peak current in differential pulse stripping voltammetry (DPSV), under optimized conditions of −0.8 V deposition potential, 5.5 pH value, 240 s deposition time, performed sensitive detection of Pb${}^{2+}$ within sensitivity of 1.811 μA · μg^−1^ and detection limit of 0.08 μg · L^−1^. Meanwhile, the results of the system in detecting lead ions in real seawater samples are highly similar to that of inductively coupled plasma emission spectrometer (ICP-MS), which proved a practicability for the system in detection of trace-level Pb${}^{2+}$.

## 1. Introduction

Currently, excessive heavy metal ions produced by industrial and domestic sewage, exhaust emissions, etc., pose a serious threat to the marine ecosystem and are increasingly dangerous to public health and safety due to their carcinogenicity, teratogenicity and mutagenicity [1,2]. The lead ion, a toxic heavy metal ion, has a trend of bioaccumulation and a strong resistance against degradation. According to the Environmental Protection Agency (EPA), approximately 20% of human exposure to Pb${}^{2+}$ occurs through contaminated drinking water [3,4,5]; therefore, a high-sensitivity, rapid-response and easy-to-use system for lead ion detection and monitoring is in demand.

Methods for detecting heavy metal ions in seawater mainly involve inductively coupled plasma emission spectrometry (ICP-OES), atomic absorption spectrometry (AAS), etc., which are precise but inconvenient to operate and nonportable. The development of MEMS techniques and nanomaterials makes electrochemical sensors more advanced and integrated, giving them features including high sensitivity, fast analytical response, simple operating process and portability. With these advantages, electrochemical sensors have become a good solution for the detection of trace-level heavy metal lead ions in a variety of environments [6,7,8].

Among nanomaterials, MWCNTs are known to have a large specific surface area, high conductivity and high material toughness and strength, which accelerate the transmission rate of electrons and improve the sensitivity of modified electrodes [9,10,11,12]. Their unique hollow and layered structure also contributes a strong absorbability to heavy metal ions [13,14]. Moreover, CS is a type of polysaccharide that has large numbers of highly active amino (–NH_2_) and hydroxyl (–OH) groups on its molecular chain [15,16], making it easy to form complexes with heavy metal ions. Meanwhile, CS can modify MWCNTs by noncovalent bonds and form stable nanocomplexes with MWCNTs, which can effectively prevent the agglomeration of MWCNTs and improve the solubility and stability of MWCNTs in CS solution [16]. Lead (Pb${}^{2+}$) ionophore IV is used as the base for the synthesis of lead-selective electrodes, as Pb${}^{2+}$ ionophore IV can provide specific channels or binding sites for lead ions under complex environments, thus enhancing the selectivity of the electrode to lead ions and improving the anti-interference and stability of the electrode [17]. Electrodes modified by such combination of MWCNT/CS were found in some literatures, showing a good sensitivity toward different substances [18,19]. Introducing in Pb${}^{2+}$ ionophore IV may provide a possibility to further improve the sensitivity to Pb${}^{2+}$ in a complex environment.

So far, a number of researchers have working on polymeric film based modified electrodes for detection of Pb${}^{2+}$ or simultaneous detection for lead ions [20,21,22,23,24,25]. In earlier works, some different chemically modified electrodes for Pb${}^{2+}$ detection were reported. Zhou Liuzhu et al. prepared a polyrutin/AgNPs-coated GCE for the simultaneous detection of the heavy metal ions Pb${}^{2+}$ and Cd^2^+ [26]. Cecylia Wardak et al. developed a new solid-contact ion-selective electrode (ISE) to Pb${}^{2+}$ detection; this electrode was modified by a polymer membrane with a nanocomposite of carbon nanofibers and an ionic liquid 1-hexyl-3-methylimidazolium hexafluorophosphate [27]. Chunwen Chang et al. proposed a graphene functionalized self-supported boron-doped diamond (G/SBDD) electrode for Pb${}^{2+}$ electrochemical detection in seawater, the electrode was prepared by in-situ graphene modification method to prepare [28]. However, many works proposed nanomaterials consisting of transition metal oxides, some of these materials required a long time and high cost of preparation. The method proposed in this paper is less time-consuming.

In this paper, a MWCNT/CS/Pb${}^{2+}$ ionophore IV/Au electrode and a matched microcavity fabricated by the MEMS technique are presented. A MWCNT/CS/Pb${}^{2+}$ ionophore IV nanomaterial membrane was synthesized by a self-assembly method and modified on the surface of the Au working electrode of the electrode by a spin-coating method. This system is characterized by low consumption, low cost, small-scale and easy operation processes to meet the detection requirements of high sensitivity, high selectivity and low detection limits and achieve the purpose of real-time detection of lead ion concentrations in seawater.

## 2. Results and Discussion

### 2.1. Morphologies and Feasibility Investigation of the Modified Electrodes

SEM was applied to investigate the surface morphology and structure of the MWCNT (a), MWCNT/CS/lead ionophore IV nanocomposite membrane, (b) modified electrodes, as shown in Figure 1a,b. Figure 1c illustrates that the diameters of the tested materials were similar, with all having an external diameter of approximately 50 nm. The tested materials were formed to be a web-like entangled structure, which provided more sites for accumulation.

The cyclic voltammetry (CV) curve and Nyquist plot of the modified electrode were measured in 0.01 mol · L^−1^ KCl and 5 mM [Fe(CN)_6_]^3−/4−^ solution, as shown in Figure 2a,b. It was found that Fe^3^+/Fe^2^+ redox peaks appeared at 0.201 V and 0.10 V, respectively. By comparing the bare Au electrode (a), MWCNT/CS/Au (b) and MWCNT/CS/Pb${}^{2+}$ ionophore IV/Au (c), it can be found that the Fe^3^+/Fe^2^+ redox peak of MWCNT/CS/Au is significantly larger than that of bare Au, indicating that the modification of MWCNT/CS on the surface of the Au electrode is successful because it increases the specific surface area of the electrode, thus improving the electrochemical performance of the electrode. However, the Fe^3^+/Fe^2^+ redox peak of MWCNT/CS/Pb${}^{2+}$ ionophore IV/Au is slightly lower than that of MWCNT/CS/Au, as Pb${}^{2+}$ ionophore IV only provides specific channels for Pb${}^{2+}$ but occupies some positions on the surface of the MWCNTs. The electron transport rate was decreased, which indicated that the MWCNT/CS/Pb${}^{2+}$ ionophore IV nanocomposite membrane was successfully modified on the Au working electrode surface of the electrode. Meanwhile, the redox peak potential difference of the MWCNT/CS/Pb${}^{2+}$ ionophore IV/Au electrode was 0.095 V, which was smaller than 0.207 V for the bare Au electrode, showing a significant increase in reversibility.

To further demonstrate the excellent electrochemical performance of the MWCNT/CS/Pb${}^{2+}$ ionophore IV nanocomposite membrane modified on the surface of the electrode, the Randles–Sevcik equation [29] was applied to accurately calculate the electroactive area of the electrode modified with different modified materials, as shown in Equation (Equation 1): (1)$$I_{p}=2.69\times 10^{5}AD^{1/2}n^{3/2}v^{1/2}c$$
where
$I_{p}$ is the peak current in amperes;*A* is the effective area of the electrode in cm^2^;*D* is the diffusion coefficient in cm^2^ · s^−1^;*n* is the number of electrons involved in the reaction;*v* is the scan rate in V · s^−1^;*c* is the concentration in mol · cm^−3^.

If *D* = 7.6 × 10^−6^ cm^2^ · s^−1^ and *n* = 1, *c* = 10 mmol · L^−1^, the active areas of the MWCNT/CS/Pb${}^{2+}$ ionophore IV/Au and bare Au electrode are theoretically 0.0495 cm${}^{2}$ and 0.0291 cm${}^{2}$ under ideal conditions. The first one shows a 67.7% larger active area than that of bare Au, which means that the amounts of substances involved in the reaction on the surface of the electrode increased after modification. In conclusion, it effectively increases the electroactive area of Au electrodes and improves the electrochemical performance of the electrode toward lead ions.

To study the dynamic characteristics of ions on the MWCNT/CS/Pb${}^{2+}$ ionophore IV/Au electrode, cyclic voltammetry was applied in this experiment in a mixed solution of KCl and K_3_[Fe(CN)_6_] with different scanning speeds, as shown in Figure 2c. This indicates that with increasing CV scanning speed, the peak redox current of [Fe(CN)_6_]^3−/4−^ also increased.

Figure 2d shows a good linear relationship between the peak current (Ip) of [Fe(CN)_6_]^3−/4−^ and $v^{1/2}$. The relevant linear equation and correlation $R^{2}$ are shown in Equation (Equation 2), indicating that the electrochemical processes on the MWCNTs/CS/Pb${}^{2+}$ ionophore IV/Au electrode were typically diffusion-controlled processes.
(2)$$\left\{\begin{array}{cc} y_{1}=11.88x+34.31, & R^{2}=0.981\\ y_{2}=-15.74x-18.38, & R^{2}=0.986 \end{array}\right..$$

### 2.2. Optimization of Parameters

#### 2.2.1. Optimization of the Deposition Potential

Deposition potential is vital to the efficiency of deposition, and optimization can accelerate the accumulation of target metal ions and improve the accuracy of the system to detect Pb${}^{2+}$ [30,31]. In this experiment, a MWCNT/CS/Pb${}^{2+}$ ionophore IV/Au electrode was applied for optimization of the deposition potential. The deposition potential was optimized between −0.6 V and −1.1 V, as shown in Figure 3a. The Pb${}^{2+}$ deposition potential increased, and the peak current also increased until the Pb${}^{2+}$ deposition potential reached −0.8 V. Here, the peak current signal of Pb${}^{2+}$ no longer increased and reached the maximum value. However, with a continuous increase in the Pb${}^{2+}$ accumulation potential, the peak current decreased as hydrogen evolution occurred on the surface of the MWCNT/CS/Pb${}^{2+}$ ionophore IV/Au electrode, which caused the Pb${}^{2+}$ to accumulate on the surface of the electrode shed. Therefore, −0.8 V was considered the optimal deposition potential of Pb${}^{2+}$.

#### 2.2.2. Optimization of the pH Value

The influence of the pH value in Pb${}^{2+}$ detection of a MWCNT/CS/Pb${}^{2+}$ ionophore IV/Au electrode was also studied, as shown in Figure 3b. When the pH of HAc−NaAc increased from 4.0 to 5.5, the peak dissolution current of Pb${}^{2+}$ also continued to increase. Furthermore, the peak current reached its maximum value when the pH changed to 5.5, and if the pH continued to increase over this value, a decrease in the current was observed instead. A pH that is too low causes H^+^ in solution to compete with Pb${}^{2+}$ (i.e., hydrogen evolution may interfere with the deposition process) [32]. Otherwise, a pH value that is too high causes Pb${}^{2+}$ to undergo hydrolysis to form metal hydroxides and complexes, thus reducing the concentration of the heavy metal Pb${}^{2+}$ in the solution [33]. Therefore, the pH value of HAc−NaAc selected was 5.5.

#### 2.2.3. Optimization of Deposition Time

The adsorption capacity of Pb${}^{2+}$ on the MWCNT/CS/Pb${}^{2+}$ ionophore IV/Au electrode with deposition time increased until it reached saturation. Therefore, the deposition time of Pb${}^{2+}$ was optimized in the range of 60 s to 360 s, as shown in Figure 3c. During the deposition time from 0 s to 240 s, the peak current of Pb${}^{2+}$ increased linearly and sharply with deposition time, which was due to the adsorption equilibrium between Pb${}^{2+}$ in solution and the Au working electrode surface. However, when the deposition time was greater than 240 s, the peak current increased slowly within 240 s to 360 s by only 9.7% as a result of rapid surface saturation at high concentrations. Therefore, 240 s was selected as the enrichment time for the whole experiment.

#### 2.2.4. Optimization of the Concentrations of MWCNTs, CS and Pb${}^{2+}$ Ionophore IV

The concentrations of MWCNTs, CS and Pb${}^{2+}$ ionophore IV all affect the detection accuracy and sensitivity of the electrode for Pb${}^{2+}$ [34]. Therefore, the effect of MWCNT concentration (0.1 mg · mL^−1^ to 2.5 mg · mL^−1^) on the Pb${}^{2+}$ stripping peak was studied under concentration of CS and Pb${}^{2+}$ ionophore IV were 0.25% and 0.7 mg · mL^−1^, respectively, as shown in Figure 3d. The peak current value of Pb${}^{2+}$ is the largest when the concentration of MWCNTs is 1.5 mg · mL^−1^, so the concentration of MWCNTs in this paper was 1.5 mg · mL^−1^. Figure 3e shows the optimized concentration of CS studied in this experiment under the optimal concentrations of MWCNTs and Pb${}^{2+}$ ionophore IV. When the CS concentration changed from 0.1% to 0.25%, the Pb${}^{2+}$ stripping peak current gradually increased, but as the CS concentration continued to increase, the Pb${}^{2+}$ dissolution peak current decreased because of the very poor conductivity of CS and a relatively high concentration of CS inhibited electron transfer on the electrode surface [35], 0.25% was selected as the optimal concentration. The optimal concentration of Pb${}^{2+}$ ionophore IV was also studied under other components were in optimal concentrations obtained above, as shown in Figure 3f. Since Pb${}^{2+}$ ionophore IV is a nonconductive polymer, too high a concentration hinders electron transmission, while too low a concentration affects its selectivity [36]. The peak current of Pb${}^{2+}$ ionophore IV reaches its maximum when the concentration is 0.7 mg · mL^−1^. Therefore, 0.7 mg · mL^−1^ was selected as the optimal concentration.

#### 2.2.5. Orthogonal Experiment

In order to verify and determine whether the composition obtained is optimal, orthogonal experiments were conducted. As only three components were used to prepare the nanomaterial film in this study, only three different concentrations of each component needed to be taken into consideration. Here, the optimal of each and their two closest concentrations were selected (respectively, 1 mg · mL^−1^, 1.5 mg · mL^−1^, 2 mg · mL^−1^ concentration for MWCNTs, 0.5 mg · mL^−1^, 0.7 mg · mL^−1^, 0.9 mg · mL^−1^ concentration for lead ionophore IV, and 0.2%, 0.25%, 0.3% for CS were selected). Based on this, nine experiments were conducted, the results are shown below in Table 1.

The averages of peck currents were calculated under one concentration constant and another two variables (e.g., average I for MWCNTs was calculated by averaging all different peck currents obtained by membranes prepared by 1 mg · mL^−1^ concentration MWCNTs and other concentrations of CS and Pb${}^{2+}$ ionophore IV). Here, nine averages were obtained and they are presented below in Table 2.

From Table 2, it is clear that 1.5 mg · mL^−1^ concentration was the best for MWCNTs on table, as under 1.5 mg · mL^−1^ concentration, the average of peck current value was the largest compared with that of other concentrations of MWCNTs. 0.7 mg · mL^−1^ and 0.25% were the best for Pb${}^{2+}$ ionophore IV and CS, respectively. Meanwhile, by comparing the ranges of averages of peck currents of these three components, the concentration of MWCNTs had the largest effect on test result as its Range R of Averages of Peck currents was the largest, while the concentration of lead ionophore had the smallest one. Here, the final optimal compositions above of the electrode in this work were verified and determined to be 1.5 mg · mL^−1^ concentration of MWCNTs, 0.7 mg · mL^−1^ concentration of Pb${}^{2+}$ ionophore IV and 0.25% concentration of CS.

### 2.3. Detection of Pb${}^{2+}$

To better study the sensitivity of the MWCNT/CS/Pb${}^{2+}$ ionophore IV nanocomposite membrane-modified electrode, under optimal conditions, the DPSV for the bare Au electrode, MWCNT/CS/Au electrode and MWCNT/CS/Pb${}^{2+}$ ionophore IV/Au electrode on a 10 μg · L^−1^ Pb${}^{2+}$ standard solution were tested. As shown in Figure 4a, the stripping peak of the bare Au electrode and other modified electrodes appeared at −0.05. The bare Au electrode showed the smallest stripping peak, while that of the MWCNT/CS/Au electrode was significantly larger than that of the bare Au electrode because the larger specific surface area of the MWCNTs was more beneficial to the adsorption of metal ions, and CS also contributed to the accumulation of Pb${}^{2+}$. Meanwhile, the peak current of the MWCNT/CS/Pb${}^{2+}$ ionophore IV/Au electrode was the largest. This is because Pb${}^{2+}$ ionophore IV can provide a specific channel for Pb${}^{2+}$ and improve the electron transport rate on the electrode surface [36].

Furthermore, electrochemical detection of Pb${}^{2+}$ at different concentrations using the MWCNT/CS/Pb${}^{2+}$ ionophore IV/Au electrode under the optimized conditions mentioned above was performed by DPSV, and the results are shown in Figure 4b. The results indicated that there was a good linear relationship between the Pb${}^{2+}$ concentration and its peak current in the range of 1 μg · L^−1^ to 100 μg · L^−1^, and the corresponding linear regression equation is given in Equation (Equation 2). The sensitivity of the system studied in this study was 1.811 μA/(μg · L^−1^), and the correlation coefficient of the system was 0.998. The relative standard deviation (RSD) of Pb${}^{2+}$ was 4.56%. Therefore, the detection limit of Pb${}^{2+}$ in this system was 0.08 μg · L^−1^ (calculated by 3σ/m, where σ is the RSD of Pb${}^{2+}$ with a 5 μg · L^−1^ concentration and *m* is the slope of the linear equation). We compare the LOD of MWCNT/CS/Pb${}^{2+}$ ionophore IV/Au electrode with similar studies in Table 3—although it does not show the widest linear range, it has a lower detection limit.
$$i_{p}=1.811c+1.663$$
where $i_{p}$ is the peak current value in μA and *C* is the concentration of Pb${}^{2+}$ in μg · L^−1^.

### 2.4. Stability and Selectivity

To verify the stability of the proposed system, an electrode was used for the detection of Pb${}^{2+}$ in a 30 μg · L^−1^ solution continuously for 7 days. The results are shown in Figure 5a, which also illustrates that the peak current curves were basically the same. To further observe its stability, as shown in Figure 5b, the peak current changing trends were drawn for 7 consecutive days. This result indicated that the peak current value of Pb${}^{2+}$ showed a slight downward trend. The RSD at 7 days was 1.16%, and no significant difference was found. The results show that the system has good stability and can be used for online detection of lead ion concentration in seawater for a few days.

The electrode-to-electrode reproducibility of the MWCNT/CS/Pb${}^{2+}$ ionophore IV/Au electrode was tested by three different planar disk electrode chips, the test result for DPVS is illustrated in Figure 5c, the peck current values of there three electrode were 6.10 μA, 5.83 μA, 6.17 μA, respectively, under 30 μg · L^−1^ Pb${}^{2+}$, the RSDs were 4.42% and 1.14%. Moreover, The RSDs of three electrodes were also studied in the concentration range of 10–100 μg · L^−1^, the RSDs was determined to be between 1.14% and 8.48%.

In addition, to further evaluate the anti-interference performance and selectivity of the system in seawater, different interferences were added to the study, and the relations between peak current variation and interfering ions are shown in Figure 5c. We added 50-times-larger concentrations of K^+^, Cl^−^, Na^+^, SO_4_^2^−. It was found that the above concentrations of interferences had no significant effect on the peak current of Pb${}^{2+}$, and the change rate of the peak current was within 5% in all cases. Meanwhile, three major heavy metal interfering ions, i.e., Cd^2^+, Cu^2^+ and Hg^2^+, were also studied. The experimental results show that the peak current change rates of 10-times-greater concentrations of Cd^2^+, Cu^2^+ and Hg^2^+ on Pb${}^{2+}$ were less than 7%, indicating that the system studied in this paper has a certain anti-interference ability in the detection of Pb${}^{2+}$.

### 2.5. Real Sample Detection

To assess the practicability of the system, seawater samples were collected from two different locations in the Maowei Sea (a semiclosed bay located in southern China). Seawater samples were directly mixed with HAc−NaAc solution at a ratio of 1:1. Under optimized conditions, the DPSV method was adopted for determination. The test results are shown in Table 4. Compared with the results of ICP-MS, the DPSV results were found to be closely similar. In addition, the accuracy of the system detection was evaluated by a recovery experiment. The recovery rate was determined to be between 94.7% and 107%, and the RSD was approximately 5%, indicating that the system studied in this paper has good sensitivity to detect and analyze seawater and can be applied to the detection of Pb${}^{2+}$ in seawater samples.

## 3. Materials and Methods

### 3.1. Reagents and Apparatus

MWCNTs (diameter 30 nm to 50 nm, purity ≥98%wt) were purchased from Chengdu Institute of Organic Sciences, Chinese Academy of Sciences, CS (high viscosity, ≥400 MPa · s) was obtained from Sigma Aldrich (Shanghai) Trading Co., Ltd. (Shanghai, China), and Pb${}^{2+}$ ionophore IV (1053.59 g/mol) was purchased from Sigma Aldrich (Shanghai) Trading Co., Ltd. (Shanghai, China). Other reagents, including ice acetic acid (CH_3_COOH), anhydrous ethanol (CH_3_CH_2_OH), anhydrous sodium acetate (CH_3_COONa), cadmium nitrate (Cd(NO_3_)_2_), copper nitrate (Cu(NO_3_)_2_), potassium chloride (KCl), etc., were all provided by Sinopharm Chemical Reagent Beijing Co., Ltd. (Beijing, China). All the chemicals used were of analytical grade, and all ultrapure water used was deionized with a resistance of 18.25 MΩ · cm (megohm · cm). Electrochemical measurements were conducted by a CHI830D electrochemical workstation (Shanghai Chenhua Instrument, Shanghai, China) with a three-electrode chip (the working electrode was modified by MWCNT/CS/Pb${}^{2+}$ ionophore IV) in a microcavity.

### 3.2. Electrochemical Measurements

The detection system used for this work mainly consist of a planar disk electrode chip and its matched microfluidic system. Before the experiments, the detection system was constructed as follows: the front of the electrode was inserted into the microcavity, and the pin of the electrode was inserted into a 3D-printed electrode connector, whose other side was connected to the electrochemical workstation. The syringe was connected to the inlet of the microcavity by fluidic channels, and another fluidic channel connected to the outlet of the microcavity was put into a liquid waste collector. The whole system is shown as Figure 6.

Pb${}^{2+}$ was detected by DPSV. The parameters of DPSV were as follows: initial potential of −0.6 V, potential increment of 5 mV, pulse period of 0.2 s, termination potential of 0.4 V and pulse width of 0.01 s. First, a certain concentration of Pb${}^{2+}$ standard solution was added to the HAc−NaAc solution (0.1 mol · L^−1^, pH=5) and then deoxygenated by nitrogen for 10 min. Furthermore, the potentiostatic method (i-t) was applied to accumulate Pb${}^{2+}$, under a −0.8 V reduction potential for 180 s. Then, a negative potential of −0.7 V was applied for 30 s for preelectrolysis. Finally, DPSV was applied to oxidize Pb deposited on the modified membrane to Pb${}^{2+}$. Here, the stripping curve and peak were obtained. After these processes, under the condition of a potential of 0.4 V, electrochemical cleaning of the electrode was carried out by the i-t method for 300 s to remove the unoxidized Pb on the surface of the working electrode [42].

### 3.3. Electrode and Microcavity Fabrication

The electrode chip used in this study mainly consisted of an Au disk electrode with a diameter of 3 mm, a Pt counter electrode and a Ag/AgCl reference electrode in a curve shape, as shown below in Figure 7a. It was fabricated on a BT resin substrate, and the photolithographic, sputtering and lift-off processes are shown below in Figure 7b.

The microcavity used for detection in this paper was composed of an upper cover plate and a lower substrate. The upper cover plate, as shown in Figure 8a, mainly included a solution inlet, microchannel, microchamber and outlet. The substrate, as shown in Figure 8b, mainly consisted of an electrode socket with a length of 15 mm, a width of 12.7 mm and a depth of 0.96 mm. The length, width and height of the whole microcavity were 50 mm, 25 mm and 8 mm, respectively. The material for fabricating the microcavity was mainly organic glass (PMMA).

The upper cover plate and the substrate were bonded and sealed by hot bonding. In the process of hot bonding, after cleaning the upper cover sheet and lower substrate, a small amount of tetrahydrofuran solvent was daubed on the substrate surface and then bonded with the upper cover sheet in a vacuum hot-pressing bonding machine under a bonding pressure of 15 kgf, a temperature of 120 °C and a bonding time of 10 min.

### 3.4. Preparation of a MWCNT/CS/Pb${}^{2+}$ Ionophore IV Nanocomposite Membrane

Due to the strong van der Waals attractions among MWCNTs, preprocessing is needed to allow hydroxyl (–OH), carboxyl (–COOH) and carbonyl (–CO) functional groups to exist on the surface of MWCNTs to improve their dispersibility [42,43]. MWCNTs were added to 120 mL of a mixed acid solution of concentrated HNO_3_ and concentrated sulfuric acid (H_2_SO_4_) (volume ratio 1:3). The solution was ultrasonically treated for 3 h, pumped, filtered, and washed until the pH was close to 6–7. After that, the filtrated product was dried and ground to powder.

CS powder (0.2 g) was added to acetic acid solution at a concentration of 0.1 mol · L^−1^ and magnetically stirred until the CS powder was completely dissolved and the CS solution was obtained.

Ten milligrams of pretreated MWCNT powder was added into 10 mL of CS solution with a concentration of 0.2% and dispersed for 30 min under ultrasonic treatment for full fusion to obtain modified materials of the MWCNT/CS nanocomposite material. Furthermore, MWCNT/CS/Pb${}^{2+}$ ionophore IV nanocomposite membrane material was prepared under the same conditions. An amount of 10 mg of pretreated MWCNT powder and 0.5 mg of Pb${}^{2+}$ ionophore IV powder were added into 10 mL of CS solution with a concentration of 0.2% and dispersed for 30 min under ultrasonic treatment. Here, the given concentrations of MWCNTs, CS solution and Pb${}^{2+}$ ionophore IV were all optimized, the method to obtain optimized concentration of MWCNT CS solution Pb${}^{2+}$ ionophore IV is given above at Section 2.

### 3.5. Modification of the MWCNT/CS/Pb${}^{2+}$ Ionophore IV/Au Electrode

The prepared electrode was soaked in anhydrous ethanol for 3 min and then washed with deionized water to remove impurities. The electrode was placed horizontally in a spin coater, and a 5 μL dispersion was dropped on the surface of the gold working electrode. Under a rotating speed of 250 r/min for 30 min, the MWCNT/CS/Pb${}^{2+}$ ionophore IV nanocomposite membrane was modified on the surface of the working electrode by spin coating. The MWCNT/CS/Pb${}^{2+}$ ionophore IV/Au electrode can be obtained.

### 3.6. Preparation of Real Samples

Seawater samples were collected from two different locations in the Maowei Sea (a semiclosed bay located in southern China), then, seawater samples were filtered through a 0.45 μm filter. The filtered real samples were stored in glass bottoms for later use. The detection processes of Pb${}^{2+}$ in real samples are mentioned above in Section 2.

## 4. Conclusions

In this paper, a Pb${}^{2+}$ detection system including a MWCNT/CS/Pb${}^{2+}$ ionophore IV/Au nanocomposite-modified electrode and a matched microcavity was proposed. The MWCNT/CS/Pb${}^{2+}$ ionophore IV nanocomposite was successfully modified on the Au electrode fabricated by the MEMS technique. The membrane was observed to have a uniform web-like entangled structure and presented better stability and higher sensitivity in all tests. Under the optimized conditions of −0.8 V deposition potential, pH 5.5 and 240 s deposition time, this system has a high sensitivity to low Pb${}^{2+}$ concentrations in the range of 1 μg · L^−1^ to 100 μg · L^−1^. In addition, it also exhibited good stability and anti-interference with 1.16% RSD and a 7% lower peak current change rate for the 7-day tests, and the Pb${}^{2+}$ detection test results for Maowei seawater were highly similar to the results of ICP-MS. 

## Figures and Tables

**Figure 1 molecules-28-04142-f001:**
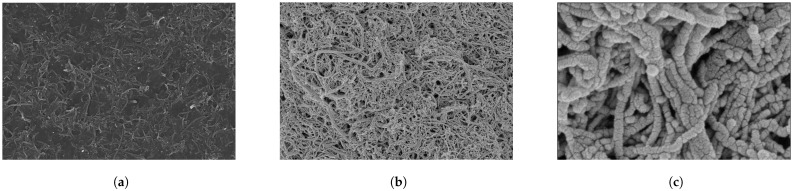
Scanning Electron Microscope (SEM) images of MWCNT (**a**), MWCNT/CS/lead ionophore IV nanocomposite material (**b**), (**c**) 100 K× magnification.

**Figure 2 molecules-28-04142-f002:**
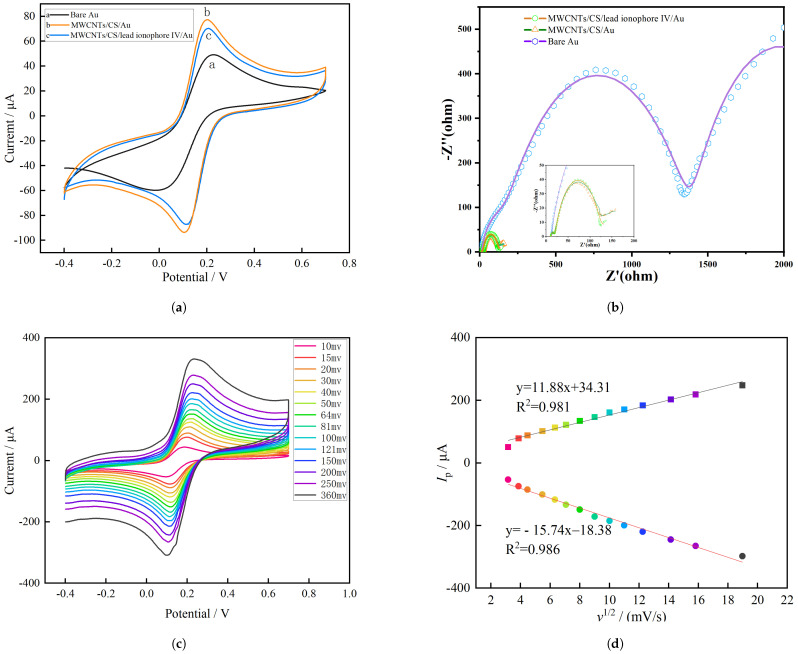
(**a**) CV of the MWCNT/CS/lead ionophore IV/Au, MWCNT/CS/Au and bare Au electrodes; (**b**) Nyquist plot of the MWCNT/CS/lead ionophore IV/Au, MWCNT/CS/Au and bare Au electrodes; (**c**) CV diagram of the MWCNT/CS/lead ionophore IV/Au under different scanning speeds; (**d**) Linear relationship between peak current $I_{p}$ and $v^{1/2}$ scanning speed.

**Figure 3 molecules-28-04142-f003:**
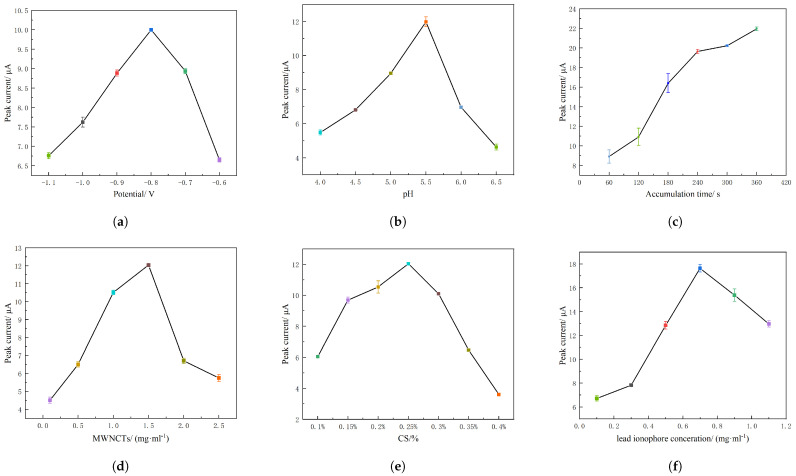
The effects of (**a**) Deposition potential, (**b**) pH value, (**c**) Deposition time, (**d**) MWNCT concentration, (**e**) CS concentration and (**f**) Pb${}^{2+}$ ionophore IV concentration on peak current.

**Figure 4 molecules-28-04142-f004:**
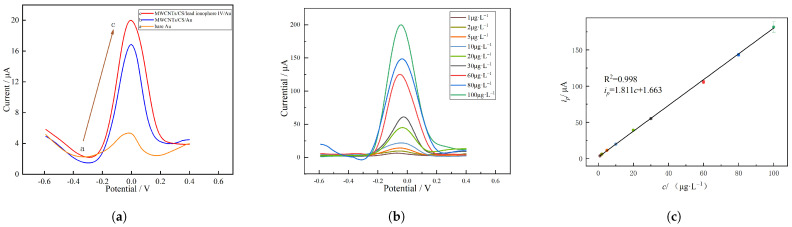
(**a**) DPSV of the bare Au electrode (curve a), MWCNT/CS/Au electrode (curve b) and MWCNT/CS/Pb${}^{2+}$ ionophore IV/Au electrode (curve c); (**b**) voltammogram of MWCNT/CS/Pb${}^{2+}$ ionophore IV/Au under different concentrations (1 μg · L^−1^ to 100 μg · L^−1^) of Pb${}^{2+}$; (**c**) calibration curve of peak current vs. Pb${}^{2+}$ concentration.

**Figure 5 molecules-28-04142-f005:**
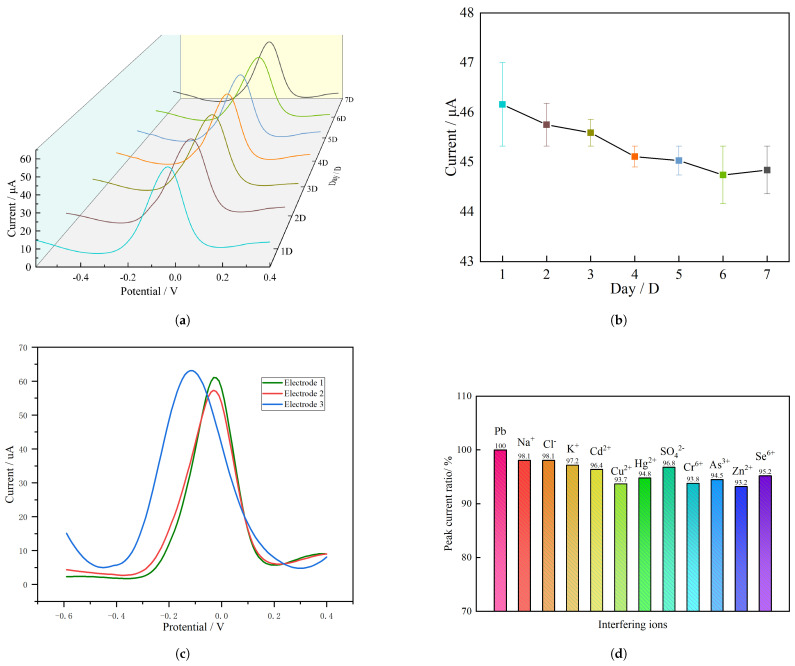
(**a**) The stripping voltammetry response of the system to Pb${}^{2+}$ for 7 consecutive days; (**b**) The peak current change rate for 7 days; (**c**) Results of DPVS for three electrodes; (**d**) The peak current difference after adding 50-times-larger concentrations K^+^, Cl^−^, Na^+^, SO_4_^2^− and 10-times-larger concentrations Cd^2^+, Cu^2^+, Hg^2^+, Cr^6^+, Zn^2^+, Se^6^+, As^3^+.

**Figure 6 molecules-28-04142-f006:**
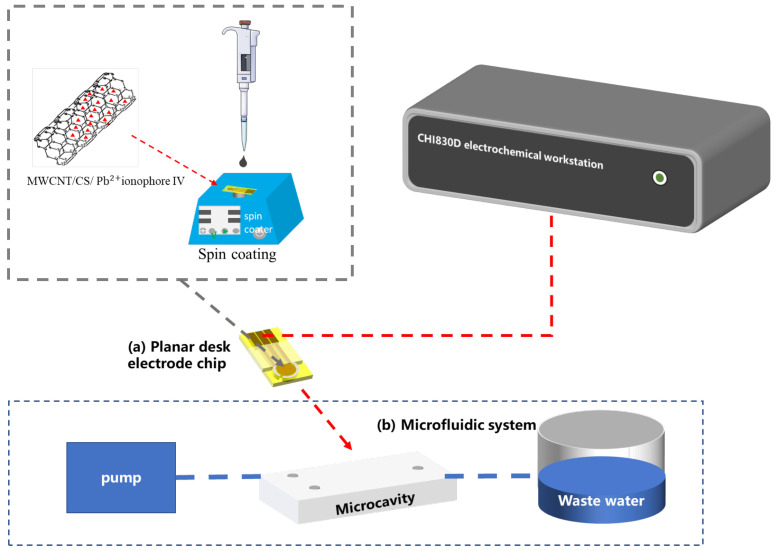
The detection system for this work is mainly composed of two parts: (**a**) a planar disk electrode chip; (**b**) a microfluidic system.

**Figure 7 molecules-28-04142-f007:**
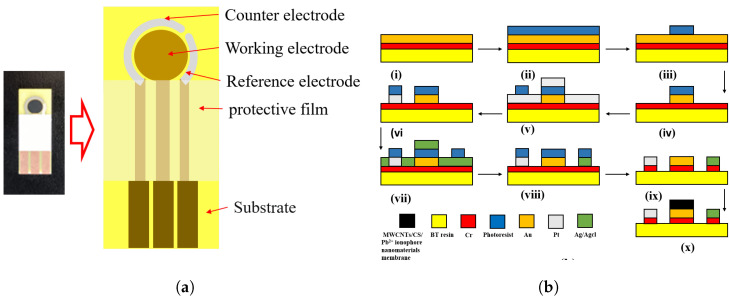
(**a**) Layout of the planar disk electrode chip; (**b**) Micro-Electro-Mechanical Systems (MEMS)-based fabrication process flow of the planar disk electrode chip.

**Figure 8 molecules-28-04142-f008:**
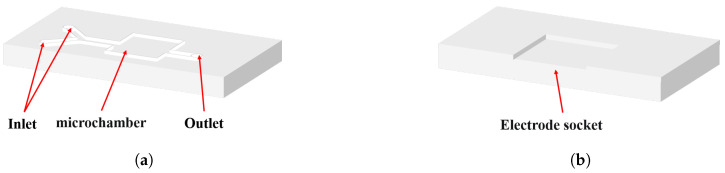
(**a**) Upper cover and (**b**) substrate design of the microcavity.

**Table 1 molecules-28-04142-t001:** Orthogonal experiment results.

Experiments	MWCNTs /(mg · mL^−1^)	Lead Ionophore IV /(mg · mL^−1^)	CS /(%)	Peck Current /(μA)
Experiment 1	1	0.5	0.2	11.090
Experiment 2	1	0.7	0.25	14.260
Experiment 3	1	0.9	0.3	13.240
Experiment 4	1.5	0.5	0.25	17.330
Experiment 5	1.5	0.7	0.3	15.770
Experiment 6	1.5	0.9	0.2	14.460
Experiment 7	2	0.5	0.3	12.740
Experiment 8	2	0.7	0.2	14.190
Experiment 9	2	0.9	0.25	15.580

**Table 2 molecules-28-04142-t002:** Average results for different concentration sets.

Average	MWCNTs	Average Peck Current	Lead Ionophore IV	Average Peck Current	CS	Average Peck Current
	/(mg · mL^−1^)	/(μA)	/(mg · mL^−1^)	/(μA)	/(%)	/(μA)
Average I	1	12.863	0.5	13.720	0.2	14.147
Average II	1.5	15.853	0.7	14.740	0.25	15.723
Average III	2	14.170	0.9	14.427	0.3	13.917
Range R	–	2.990	–	1.020	–	2.476

**Table 3 molecules-28-04142-t003:** Linear ranges and detection limits of different electrodes for Pb${}^{2+}$.

Method	Linear Range	LOD	Reference
	/(μg · L^−1^)	/(μg · L^−1^)	
Pt/ISM (9%NC)	5–10.000 μmol · L^−1^	3.1 μmol · L^−1^	[27]
G/SBDD	1–100	0.21	[28]
AuNPs-SPCE	2–500	4.4	[37]
Bi_2_O_3_@CNTs/GCE	2–40	3.4	[38]
IDB/GCE	20.7–207	3.5	[39]
PPy/NH_2_-MIL-53/GCE	1–400	0.31	[40]
UIO-66-NH_2_/GaOOH/GCE	114–517	5.8	[41]
MWCNT/CS/Pb${}^{2+}$/Au	1–100	0.08	this work

**Table 4 molecules-28-04142-t004:** Detection of Pb${}^{2+}$ concentration in real samples.

Seawater	Found	ICP-MS	Added	Found	Recovery	RSD (%)
Samples	/(μg · L${}^{-1}$)	/(μg · L${}^{-1}$)	/(μg · L${}^{-1}$)	/(μg · L${}^{-1}$)	/(%)	(*n* = 3)
	1.86 ± 0.13	1.8	0	–	–	–
Sample 1	–	–	5	6.70 ± 0.20	96.8%	3.1%
	–	–	10	11.33 ± 0.50	94.7%	4.4%
	1.14 ± 0.08	1.2	–	–	–	–
Sample 2	–	–	5	6.49 ± 0.29	107%	4.5%
	–	–	10	10.85 ± 0.56	97.1%	5.1%

## Data Availability

The data that support the findings of this study are available from the corresponding author upon reasonable request.

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
