# Peer review of "A Planar Disk Electrode Chip Based on MWCNT/CS/Pb2+ Ionophore IV Nanomaterial Membrane for Trace Level Pb2+ Detection"

_molecules, 2023, doi:10.3390/molecules28104142_

Round 1

Reviewer 1 Report

Zhuang et al developed a multiwalled carbon nanotube (MWCNTs)/ chitosan (CS)/ lead (Pb2+) ionophore IV modified Au microelectrode used to detect Pb2+. The following items should be addressed before its publication in this Journal.  

1. So far, the electrochemical sensors to the detection of Pb2+ have developed very well by some authors. In introduction, authors should introduce other electrochemical sensors to the detection of Pb2+ briefly.

2. If it is supposed to be a sensor paper, it needs substantially improved multiple measurements of the calibration curve with error bars based on at least three, preferably more, repetitions, for example, in Figures 6 and 7.

3. The electrode-to-electrode reproducibility at relevant concentrations should be reported.

4. In selectivity experiment, these heavy metal ions should be tested, such as Cr6+, Te3+Co3+Se6+, As3+ etc.

5. The obtained LOD values should be compared with other results reported by electrochemical sensors in the literature.

The mechanism of the detection of Pb2+ at the proposed sensor should be discussed in the paper. 

Author Response

Dear editors and reviewers:

On behalf of all the contributing authors, I would like to express our sincere appreciations to all editors and reviewers, especially your constructive comments concerning our article originally entitled “A microelectrode based on MWCNT/CS/Pb2+ ionophore IV nanomaterial membrane for trace level Pb2+ detection” (newly renamed to “A planar disk electrode chip based on MWCNT/CS/Pb2+ ionophore IV nanomaterial membrane for trace level Pb2+ detection, Manuscript ID: molecules-2381268). These comments are all valuable and helpful for improving our article. According to the associate editor and reviewers’ comments, we have made extensive modifications and descriptions to make our results more convincing, some of main revised parts are listed below. In this revised version, changes to our manuscript were all highlighted within the document by using red-colored text. Point-by-point responses to the nice associate editor and four nice reviewers are listed below this letter. Sincerely thanks again.

Main revised parts:

(1) Title of manuscript was changed

(2) ‘Results and discussion’ was resectioned to Section 2 and ‘Materials and Methods’ to Section 3 according to molecules layout.

(3) Two subsections (Orthogonal experiment for determination and verification of final optimal composition and preparation of real samples) and a brief literature review were added in.

(4) Fig 2, 3, 4, 5, 6, 7, 8 were revised, and more details were given in main content following review’s comments.

Best regards to all editors and reviewers

Qiu Chenjun

E-mail: qiuchengjun@bbgu.edu.cn

Response to reviewer 1

1. So far, the electrochemical sensors to the detection of Pb2+ have developed very well by some authors. In introduction, authors should introduce other electrochemical sensors to the detection of Pb2+ briefly.

Thanks for a constructive comment here to improve our manuscript. A brief literature review that introduces other electrochemical sensors is added into the introduction section, highlighted with red text, please check on the new manuscript.

2.If it is supposed to be a sensor paper, it needs substantially improved multiple measurements of the calibration curve with error bars based on at least three, preferably more, repetitions, for example, in Figures 6 and 7.

We sincerely appreciate reviewer as all these suggestions are valuable and precise, also provide us a direction to make our manuscript better. In our new submission, figures 6 and 7 were revised followed this comment, (figures 6 and 7 have been changed to fig 2 and fig 3 based on layout of molecules).

 3. The electrode-to-electrode reproducibility at relevant concentrations should be reported.

Thanks for this constructive comment that help us to improve, we have added a supplement that describe the electrode-to-electrode reproducibility in the concentration range of 10−100 μg · L−1 lead ion solution, the RSDs calculated are reported at section 2.4 in the new manuscript, the supplement is marked in red.

4. In selectivity experiment, these heavy metal ions should be tested, such as Cr6+, Te3+Co3+Se6+, As3+ etc.

Thanks for comment, a new graph that supplements Cr6+, Zn2+, Se6+, As3+ was added in to replace the last one, please check on fig 5d. We are sorry that we could not test the selectively against Te3+、Co3+, because we have not prepared the stander solution for Te3+、Co3+ in lab, and it is also too difficult to purchase these within 5 revision days. We will take this comment as a note to improve our future study.

5. The obtained LOD values should be compared with other results reported by electrochemical sensors in the literature.

Thanks for a constructive comment here, we have supplemented a table to compare LOD and linear range of different electrochemical sensors on lead ion detection, please check on section 2.3, table 3.

Reviewer 2 Report

1.      Why do the Authors call the electrode a microelectrode? This is unjustified, as the dimensions of the working electrode (3 mm in diameter) significantly exceed 25 micrometers.

2.      Part 2.3. Preparation of a MWCNT/CS/Pb2+ ionophore IV nanocomposite membrane and part 2.4. Modification of the MWCNT/CS/Pb2+ ionophore IV/Au microelectrode should be rewritten in such a way as to take into account the optimization of the electrode composition. How was MWCNT/CS/Au electrode obtained?

3.     Page 5 lines 135-136  Authors wrote “The cyclic voltammetry (CV) curve and Nyquist plot of the modified electrode were measured in 0.01 mol · L1 KCl and 5 mM [Fe(CN)6]3 – /4 – solution, as shown in Figure 5(a)” In Fig 5(a) only CV curves are shown. There is no Nyquist plot.

4.      Line 159 Are the values of the active area of the electrodes given correctly? Is the area of MWCNT/CS/Pb2+ ionophore IV/Au,electrode larger than that of MWCNT/CS/Au electrode?

5.      Fig. 5. Figures 5(b) and 5(c) have different designations inside Figures a and b, respectively. This is confusing and completely redundant.

6.      Fig. 7. Figures 7(b) and 7(c) have different designations inside Figures a and b, respectively. This is confusing and completely redundant.

7.      Subsection 3.2 is unclear. How was the optimization done? For which electrode was the optimization of potential and accumulation time and pH carried out?

What were the concentrations of the other ingredients when optimizing the concentration of the electrode modifying ingredients? e.g. What concentrations of CS and Pb2+ ionophore IV were used during the optimization of MWCNTS content in the range of 0.1-2.5 mg/mL?

What was the final optimal composition of the electrode selected?

8.      Real sample preparation is omitted.

9.      Lead ionophore is primarily used in potentiometric electrodes. It would be very interesting to compare the parameters of the electrode described in this work with the parameters of other electrodes using this ionophore e.g. (Materials 16 (2023) 1003; Desalination and Water Treatment 51 (2013) 658-664; and other.

Author Response

Dear editors and reviewers:

On behalf of all the contributing authors, I would like to express our sincere appreciations to all editors and reviewers, especially your constructive comments concerning our article originally entitled “A microelectrode based on MWCNT/CS/Pb2+ ionophore IV nanomaterial membrane for trace level Pb2+ detection” (newly renamed to “A planar disk electrode chip based on MWCNT/CS/Pb2+ ionophore IV nanomaterial membrane for trace level Pb2+ detection, Manuscript ID: molecules-2381268). These comments are all valuable and helpful for improving our article. According to the associate editor and reviewers’ comments, we have made extensive modifications and descriptions to make our results more convincing, some of main revised parts are listed below. In this revised version, changes to our manuscript were all highlighted within the document by using red-colored text. Point-by-point responses to the nice associate editor and four nice reviewers are listed below this letter. Sincerely thanks again.

Main revised parts:

(1) Title of manuscript was changed

(2) ‘Results and discussion’ was resectioned to Section 2 and ‘Materials and Methods’ to Section 3 according to molecules layout.

(3) Two subsections (Orthogonal experiment for determination and verification of final optimal composition and preparation of real samples) and a brief literature review were added in.

(4) Fig 2, 3, 4, 5, 6, 7, 8 were revised, and more details were given in main content following review’s comments.

Best regards to all editors and reviewers

Qiu Chenjun

E-mail: qiuchengjun@bbgu.edu.cn

Response to reviewer 2

1. Why do the Authors call the electrode a microelectrode? This is unjustified, as the dimensions of the working electrode (3 mm in diameter) significantly exceed 25 micrometers.

We are so appreciated that reviewer points out our mistake here, we have revised “microelectrode” to “a planar disk electrode chip” in our new submission.

2. Part 2.3. Preparation of a MWCNT/CS/Pb2+ ionophore IV nanocomposite membrane and part 2.4. Modification of the MWCNT/CS/Pb2+ ionophore IV/Au microelectrode should be rewritten in such a way as to take into account the optimization of the electrode composition. How was MWCNT/CS/Au electrode obtained?

Thanks for a comment that helps us to improve our paper, according to the molecules layout, we have resectioned 'Results and discussion' to Section 2 and 'Materials and Methods' to Section 3, which means the optimization of the electrode composition will be introduced first. Otherwise, this part was also revised by adding description of preparation of MWCNT/CS/Au electrode and selection of concentration of electrode composition, marked in red.

3. Page 5 lines 135-136 Authors wrote “The cyclic voltammetry (CV) curve and Nyquist plot of the modified electrode were measured in 0.01 mol · L1 KCl and 5 mM [Fe(CN)6]3 – /4 – solution, as shown in Figure 5(a)” In Fig 5(a) only CV curves are shown. There is no Nyquist plot.

Thanks to point out our careless mistake here, the Nyquist plot was supplemented in our new manuscript, please check on section 2.1, fig 2b.

4. Line 159 Are the values of the active area of the electrodes given correctly? Is the area of MWCNT/CS/Pb2+ ionophore IV/Au,electrode larger than that of MWCNT/CS/Au electrode?

Thanks for this question. The values of active area of the electrodes in this paper were actually obtained by theoretical calculation based on ideal conditions, these must be different with the actual values of that. The purpose we apply these values here was just to show a large difference between active area of MWCNT/CS/Pb2+ ionophore IV/Au electrode and bare Au electrode, then to indicate a possible factor that make MWCNT/CS/Pb2+ ionophore IV/Au electrode better on lead detection than that of bare Au. In our new manuscript, we have deleted the value of MWCNT/CS/Au electrode.

5. Fig. 5. Figures 5(b) and 5(c) have different designations inside Figures a and b, respectively. This is confusing and completely redundant.

We are sorry about this careless mistake, we have corrected it in our new submission, please check this on fig2c, 2d.

6. Fig. 7. Figures 7(b) and 7(c) have different designations inside Figures a and b, respectively. This is confusing and completely redundant.

We are sorry about this careless mistake, we have corrected it in our new submission, please check this on fig4b, 4c.

7. Subsection 3.2 is unclear. How was the optimization done? For which electrode was the optimization of potential and accumulation time and pH carried out?

What were the concentrations of the other ingredients when optimizing the concentration of the electrode modifying ingredients? e.g. What concentrations of CS and Pb2+ ionophore IV were used during the optimization of MWCNTS content in the range of 0.1-2.5 mg/mL?

What was the final optimal composition of the electrode selected?

Thanks for this question. In our new manuscript, we have made a supplement for this question as well as the final optimal composition, please check on 2.2, marked in red. Otherwise, we have also added a new subsection for an orthogonal experiment that verified the final composition used in this work is optimal, please check on 2.3, thanks again for a question that help us to improve.

8. Real sample preparation is omitted.

For this question, we have added a new subsection (3.6) to describe how the real samples were prepared.

9. Lead ionophore is primarily used in potentiometric electrodes. It would be very interesting to compare the parameters of the electrode described in this work with the parameters of other electrodes using this ionophore e.g. (Materials 16 (2023) 1003; Desalination and Water Treatment 51 (2013) 658-664; and other.

Thanks for a constructive comment and recommendation here. We have checked these literatures carefully and added these as references. Meanwhile, we have supplemented the comparison of different electrode for lead ion detection, please check on section 2.3, table 3. These two papers are quite good studies so we have added these into literature review and comparation table.